# In Vivo Detection of Circulating Cancer-Associated Fibroblasts in Breast Tumor Mouse Xenograft: Impact of Tumor Stroma and Chemotherapy

**DOI:** 10.3390/cancers15041127

**Published:** 2023-02-09

**Authors:** Tao Lu, Lisa Oomens, Leon W. M. M. Terstappen, Jai Prakash

**Affiliations:** 1Engineered Therapeutics, Department of Advanced Organ Bioengineering and Therapeutics, TechMed Centre, Faculty of Science and Technology, University of Twente, Drienerlolaan 5, 7500 AE Enschede, The Netherlands; 2VyCAP B.V., Capitool 41, 7521 PL Enschede, The Netherlands; 3Medical Cell BioPhysics, Faculty of Science and Technology, University of Twente, Hallenweg 23, 7522 NH Enschede, The Netherlands

**Keywords:** circulating tumor cells, circulating cancer-associated fibroblasts, metastasis, tumor microenvironment, tumor stroma

## Abstract

**Simple Summary:**

Cancer-associated fibroblasts (CAFs) play an important role in tumor progression. They can circulate with tumor cells to support the survival and formation of metastasis. The aim of our study was to investigate circulating CAFs and CTCs in breast cancer model to understand the effect of tumor stroma and chemotherapy on cCAFs and CTCs formation. We found that tumors with CAFs showed faster growth than tumors without CAFs, as well as more cCAFs detected. Besides, ITGA5 as a new biomarker for CAFs showed good agreement with the markers FAP and α-SMA. Furthermore, liposomal doxorubicin suppressed tumor growth, however, more cCAFs and CTCs were detected in the chemotherapy-treated mice. These findings suggest that tumor stroma benefits tumor growth and chemotherapy may increase the formation of cCAFs and CTCs, potentially promoting tumor metastases.

**Abstract:**

Cancer-associated fibroblasts (CAFs) are important drivers in the tumor microenvironment and facilitate the growth and survival of tumor cells, as well as metastasis formation. They may travel together with tumor cells to support their survival and aid in the formation of a metastatic niche. In this study, we aimed to study circulating CAFs (cCAFs) and circulating tumor cells (CTCs) in a preclinical breast tumor model in mice in order to understand the effect of chemotherapy on cCAFs and CTC formation. Tumors with MDA-MB-231 human breast tumor cells with/without primary human mammary fibroblasts (representing CAFs) were coinjected in SCID mice to develop tumors. We found that the tumors with CAFs grew faster than tumors without CAFs. To study the effect of the stroma on CTCs and cCAFs, we isolated cells using microsieve filtration technology and established ITGA5 as a new cCAF biomarker, which showed good agreement with the CAF markers FAP and α-SMA. We found that ITGA5+ cCAFs shed in the blood of mice bearing stroma-rich coinjection-based tumors, while there was no difference in CTC formation. Although treatment with liposomal doxorubicin reduced tumor growth, it increased the numbers of both cCAFs and CTCs in blood. Moreover, cCAFs and CTCs were found to form clusters in the chemotherapy-treated mice. Altogether, these findings indicate that the tumor stroma supports tumor growth and the formation of cCAFs. Furthermore, chemotherapy may exacerbate the formation of cCAFs and CTCs, which may eventually support the formation of a metastasis niche in breast cancer.

## 1. Introduction

Breast cancer is one of the most common malignant cancers worldwide [1]. Around one-third of breast cancer patients will develop metastasis, which is the main cause of cancer morbidity and mortality [2,3]. The metastasis of cancer involves a series of sequential and interrelated steps; cancer cells detach from the primary tumor; intravasate into the circulatory and lymphatic systems, termed circulating tumor cells (CTCs); extravasate to distant capillary beds; and invade and proliferate in distant organs [4]. Evidence shows that the presence of CTCs in circulation strongly correlates with the prognosis and survival of cancer patients [5,6,7,8,9,10]. A study on non-small-cell lung cancer indicated that a much higher number of CTCs was detected in patients with stage IV lung cancer than in those with stage III lung cancer, showing poor progression-free and overall survival in patients with more CTCs [7]. Another study showed that the presence of ≥ 5 CTCs was associated with poor progression-free survival and overall survival in metastatic breast cancer patients [11]. A similar outcome was reported in metastatic colorectal cancer, showing lower overall survival in patients with more CTCs in their blood [8].

Apart from CTCs, a few studies have demonstrated that metastatic tumor cells can migrate along with the stromal components including cancer-associated fibroblasts (CAFs) from the primary sites—so-called circulating CAFs (cCAFs) [12,13]. In the tumor microenvironment, CAFs are programmed by different factors such as cytokines and growth factors secreted by cancer cells. These CAFs are the most abundant stromal cells in the tumor microenvironment and are believed to help cancer cells to intravasate into the bloodstream, support the survival of CTCs during circulation, extravasate and grow in distant sites [14,15]. Therefore, detection of CTCs and cCAFs is of high importance to monitor the progression of the disease, as well as the effectiveness of therapy and guide treatment [16].

Successful CTC detection has been reported in several cancer types, including lung, breast, prostate and colorectal cancer, as well as gastrointestinal and biliary cancers [17]. Because most cancer cells originate from epithelial cells, a couple of epithelial biomarkers are exclusively expressed in cancer cells but not in other non-cancer cells in the blood, which facilitates the detection of CTCs from blood samples [17]. For instance, EpCAM (epithelial cell adhesion molecule) and pan-CK (pan-cytokeratin) have been widely used as biomarkers in CTC capture and identification in many studies [18,19,20,21,22,23]. Unlike CTCs, CAFs have multiple sources of origin. Several types of cells in tumor tissue can transdifferentiate into CAFs, vascular smooth muscle cells, pericytes, bone marrow cells and endothelial cells, which suggests heterogeneity in CAFs, leading to possible difficulty in the use of a specific marker to detect all CAF populations [24]. Biomarkers have been reported to have the potential to identify CAFs, commonly including α-smooth muscle actin (α-SMA), fibroblast-specific protein, vimentin and fibroblast activation protein (FAP) [25].

In our previous study, we showed that integrin α5 (ITGA-5), a receptor of fibronectin, is overexpressed in pancreatic CAFs [26]. In the current study, we hypothesized that cCAFs might retain the expression of ITGA5 after their shedding from the primary tumor and therefore might serve as a novel biomarker to identify cCAFs.

## 2. Materials and Methods

### 2.1. Cell Culture

An MDA-MB-231 human breast cancer cell line was cultured in high-glucose DMEM with L-glutamine (GE Healthcare) with 10% fetal bovine serum (FBS) medium. Human mammary fibroblast (HMF) primary cells were obtained from ScienCell and cultured in FM (ScienCell) with 2% FBS, 1% fibroblast growth supplement and 1% penicillin/streptomycin. All cells were cultured in a cell incubator at 37 °C with 5% CO_2_.

### 2.2. Quantitative Polymerase Chain Reaction (qPCR)

A total of 40,000 cells/well were seeded in a 12-well plate overnight and were starved by starvation medium (culture medium without FBS) for 24 h, followed by TGFβ treatment (5 ng/mL in serum-free medium) or conditioned medium (serum-free) from corresponding tumor cells for 24 h. To collect tumor-conditioned medium, tumor cells were cultured in a T75 flask, reaching ~90% confluence in normal medium, which was then replaced with 8 mL fresh serum-free medium for 24 h. Subsequently, the conditioned medium was collected and frozen at −20 °C before use.

Total RNA was isolated using a GenElute Mammalian Total RNA Miniprep Kit (Sigma, St. Louis, MI, USA), and the amount of RNA was measured with a NanoDrop ND-1000 spectrophotometer (Thermo Scientific, Waltham, MA, USA), followed by cDNA synthesis with an iScript cDNA synthesis kit (Bio-Rad, Hercules, CA, USA). A total of 10 ng cDNA was used for each PCR reaction. The PCR primers were purchased from Sigma. Gene expression levels were normalized to the expression of the housekeeping gene.

### 2.3. Immunofluorescence Staining

FAP Alexa-647 conjugated antibody (Clone# 427819), ITGA 5 Alexa-488 conjugated antibody (Clone# 238307) and vimentin Alexa-488 conjugated antibody (Clone# 280618) were obtained from R&D Systems. α-SMA Alexa-488 conjugated antibody was obtained from Thermo Fisher (Clone# 1A4). Pan-cytokeratin PE-conjugated antibody was obtained from Cell Signaling Technology (Clone# C-11), and pan-cytokeratin eFluor 570 conjugated antibody was purchased from Thermo Fisher (Clone# AE1/AE3). Hoechst dye was obtained from Thermo Fisher. These labeled antibodies and dyes were diluted to appropriate concentrations before use (Appendix A).

A total of 5000 cells/well were seeded in a 48-well plate with a similar treatment as that mentioned for qPCR analysis procedures. After 48 h activation with TGFβ, cells were fixed with 4% formaldehyde (Sigma) at room temperature for 10 min. After washing 2 times with phosphate-buffered solution (PBS), 0.1% Triton permeabilization agent was added to cells for 10 min at room temperature, followed by one cycle of PBS washing. Fluorescently labeled antibodies and Hoechst dye were diluted in 1% BSA at an appropriate concentration (Appendix A) and coincubated with fixed cells for 1 h in a dark environment at room temperature. Cells were washed 2 times in PBS before imaging. Subsequently, imaging was performed using Nikon Ti-E automated inverted fluorescence microscope at a magnification of 20× and an exposure time of 50 ms for the DAPI filter cube, 250 ms for the FITC filter cube, 250 ms for the PE filter cube and 500 ms for the APC filter cube.

### 2.4. Preparation of Liposomal Doxorubicin

Liposomal doxorubicin (L-Dox) was prepared as previously reported [27]. Briefly, L-Dox was prepared with HSPC/cholesterol/DSPE-PEG at a molar ratio of 5.5/4/0.5, respectively, using thin lipid film hydration in ammonium sulfate (250 mM), followed by heated extrusion [28]. Small unilamellar vesicles were obtained by extrusion through Nuclepore^®^ (Whatman Inc., Clifton, NJ, USA) filters with pore sizes ranging from 200 to 100 nm on a Thermo barrel extruder at 65 °C. Doxorubicin was loaded based on an ion gradient method called remote loading [28,29]. The diameter and polydispersity index (PDI) were measured using a Zetasizer nano-ZS (Malvern Instruments Ltd., Malvern, UK). L-Dox with size of approximately 100 nm and a PDI below 0.1 was used in this study.

### 2.5. Animal Studies

All animal protocols and procedures for this study were approved by the Institutional Animal Care and Use Committee in Twente University. Female SCID mice aged 5–7 weeks were obtained from Janvier. Mice were divided into the 3 groups as follows: Mice in groups 1 and 2 were orthotopically injected with 2 million MDA-MB-231 cells and 2 million HMFs mixed in Matrigel (1:1 *v*/*v*); group 3 mice were orthotopically seeded with 2 million MDA-MB-231cells only. L-Dox was intravenously injected in group 1 mice (tumor formed with tumor cells and HMFs) when the tumor size reached ~400 mm^3^ at a dosage of 5 mg/kg per week for three consecutive weeks. The remaining groups of mice were treated with PBS at corresponding time points. Blood was collected when group 2 mice reached a tumor volume of 1000 mm^3^ or the humane endpoint.

### 2.6. Blood Collection and Microsieve Filtration Processing

Blood was taken from mice in EDTA tubes, followed by 1% PFA fixation overnight. Then, the blood samples were filtrated through a microsieve platform with a chip pore size of 5 µm (Microsieve, VyCAP) [23]. A similar fixation and permeabilization process was performed, and samples were coincubated with corresponding fluorescently labeled antibodies as mentioned in the Section 2.3, followed by scanning under a Nikon Ti-E automated inverted fluorescence microscope.

## 3. Results

### 3.1. Selection of Biomarkers for Cancer-Associated Fibroblasts and Cancer Cell Identification

In this study, we first examined several reported biomarkers for CAFs on HMFs to characterize whether they also express these markers (Figure 1). Besides well-known markers such as α-SMA and FAP, we also examined ITGA5, which we previously found to be overexpressed on pancreatic CAFs [26]. At the mRNA level, we found that α-SMA and FAP, as well as ITGA5, were significantly upregulated in HMFs activated by TGFβ or tumor cell conditioned medium compared to non-activated HMFs and cancer cells, in which these markers were absent (Figure 1A).

Furthermore, we performed in vitro cell staining, which showed induced ITGA5 and FAP expression levels in activated HMFs compared to non-activated HMF or cancer cells (Figure 1B,C). α-SMA cell staining did not reveal differences between activated and non-activated HMFs, which is likely due to the already activated state of the HMFs or the failure of the fluorophore-conjugated antibody. Therefore, we tested a different α-SMA unconjugated antibody in a Western blot assay, showing increased protein expression of α-SMA after activation by TGFβ or cancer cell conditioned medium (Appendix A). Vimentin failed to show differences between HMFs and cancer cells, regardless of TGFβ stimulation, at both the gene and protein levels, indicating non-specificity (Appendix A). We confirmed that pan-cytokeratin (pan-CK), a widely used biomarker for CTC identification [23], is specifically expressed in MDA-MB-231 breast cancer cells without any signal in HMFs (Figure 1B). Hence, we selected FAP and ITGA5 as biomarkers to detect CAFs and pan-CK as a marker of tumor cells in the following experiment.

### 3.2. Identification of CAFs and Tumor Cells Using a Microsieve Platform

To isolate and identify single cells, we used a cell-size-based microsieve filtration system and enriched cCAFs and CTCs. Figure 2A shows a schematic illustration of the workflow used to isolate single cells. In brief, blood collected from mice was added to the microsieve, and cells of interest were enriched by microsieve filtration, followed by incubation with fluorophore-labeled antibody; subsequently, images were captured using a fluorescent microscope for enumeration.

To set up the workflow, activated HMFs (representing CAFs) and MDA-MB-231 breast cancer cells were mixed and processed through the microsieve system, followed by fluorophore-conjugated antibody of FAP, ITGA5 and pan-CK incubation. All FAP-positive cells (activated HMFs) showed clear costaining with ITGA5 (yellow arrows in Figure 2B). In contrast, unique staining of pan-CK was observed in MDA-MB-231 cells (red arrows in Figure 2B), which did not show any signal for ITGA5. These results suggest that ITGA5 can be used as a new biomarker to detect cCAFs. We also mixed these cells with mouse blood and tested in them in the microsieve platform. However, the applied FAP antibody resulted in strong unspecific binding in mouse blood components (Appendix A). Importantly, no unspecific binding of ITGA 5 and pan-CK antibodies was observed in the mouse blood; therefore, we kept ITGA5 and pan-CK for the following blood sample analysis.

### 3.3. CAFs Promoting Tumor Growth and Supporting Dissemination of CTCs and cCAFs

Having demonstrated that ITGA5 and pan-CK can be used for CAF and tumor cell detection, we employed the microsieve filtration system to capture cCAFs and CTCs in blood from orthotopic MDA-MB-231 breast-tumor-bearing mice. The mice were divided into two groups i.e., (1) those with tumor cells alone (T alone) and (2) those with both tumor cells and fibroblasts (HMFs) (T + F). cCAFs were detected in both groups of mice (Figure 3A), among which with more cCAFs captured from T + F mice compared to T-alone mice, although cCAFs were detected in four of six mice (66.7%) in both groups. Two of six mice (33.3%) presented with more than five cCAFs per 100 μL of blood in the T + F group. In comparison, the T-alone group was found have few cCAF (maximum of 1.5 CAFs/100 μL of blood; Figure 3B). Similarly, more CTCs were detected in T + F mice (2/6, 33.3%), showing one to two CTCs/100 μL of blood, while only one mouse in the T-alone group had CTCs and at very low numbers (0.5 CTCs/100 μL blood) (Figure 3A,B). Furthermore, significantly rapid tumor growth was observed in mice injected with tumor cells + fibroblasts compared to mice with tumor cells alone (*p* = 0.0169, *p* = 0.0424), likely due to the protumoral effect of CAFs (Figure 3C).

### 3.4. Chemotherapy Inducing More cCAFs and CTCs in Mice with Breast Tumors

Liposomal doxorubicin (L-Dox) is a classical chemotherapeutic drug approved by the FDA as Doxil™ to treat several cancers, including breast cancer.

Mice were administrated L-Dox once per week for three consecutive weeks. A clear delay in tumor growth was observed after three weeks of chemotherapy treatment, resulting in a significantly smaller tumor weight in comparison to mice treated with PBS (Figure 4A). However, more CTCs and a higher incidence of CTC detection were observed in mice treated with L-Dox compared to control mice (Figure 4B,C). CTCs were detected in the blood of six out of six L-Dox-treated mice (100%), among which two mice (33.3%) showed more than 100 CTCs per 100 μL of blood (max. 160 CTCs/100 μL). In comparison, CTCs failed to be detected in four out of six mice (66.7%) in the non-chemotherapy group. Only two mice in non-chemotherapy group were CTC-positive but with very low numbers (<2 CTCs/100 μL blood) (Figure 4D).

Similar results were reported in the case of cCAF detection. A high cCAF detection incidence and cCAF numbers were observed in L-Dox-treated mice, with six out of six mice positive for cCAFs and two mice were detected with over 100 cCAFs per 100 μL blood (max. 600 cCAFs/100 μL; Figure 5A–C). cCAFs were detected in four out of six non-chemotherapy-treated mice but with low numbers (max. 14.3 cCAFs/100 μL) (Figure 5C). Interestingly, we found that cCAFs were accompanied by CTCs, especially in L-Dox-treated mice (Figure 4D and Figure 5C).

### 3.5. Detection of cCAFs and CTC Clusters in Blood from L-Dox-treated Mice with Breast Tumors

Besides the single CTCs and CAFs captured by our microsieve system, we also identified homotypic clusters of circulating CAFs and cancer cells in mouse blood (Figure 6A), as well as heterotypic cCAF–CTC clusters (Figure 6B).

However, clusters of cCAFs and CTCs were detected only in two L-Dox-treated mice. A total of 18 cCAF clusters and 17 CTC clusters were captured in mouse No. 2, and mouse No. 5 showed 68 cCAF clusters and 3 CTC clusters in the blood. In comparison, no cluster of cCAFs or CTCs was detected in other groups of mice, regardless of whether the tumor seeded alone or with fibroblasts (Figure 6C). It is worth mentioning that these two cluster-positive mice also had much higher numbers of single CTCs and cCAFs in blood as compared to other mice (Figure 4D and Figure 5C).

## 4. Discussion

The present study unravels a number of findings in relation to the role of the tumor stroma and chemotherapy in the field of CTCs and circulating stromal cells. These findings include that (1) CAFs contribute to the tumor growth of breast tumors, as seen in our preclinical study; (2) CAFs, as well as CTCs, shed in blood circulation; (3) ITGA5 can be considered a new marker for cCAFs; (4) treatment with chemotherapy (L-Dox) induced increased incidence and numbers of cCAFs and CTCs; and (5) cCAFs and CTCs were found to form clusters in chemotherapy-treated mice. These findings signify the importance of the tumor stroma in the formation of cCAFs, which might eventually help in creating metastatic niches in other organs. Importantly, our findings that chemotherapy enhanced the formation of CTCs and cCAFs sheds new light on the fact that chemotherapy exacerbates metastasis in breast cancer, as reported in other studies [30,31,32]. These results are not only important for further investigations to understand the role of tumor stroma in preclinical studies but also in patient samples.

A growing number of studies have focused on CTC analysis of clinical samples to predict metastasis [6,7,8,33]. However, few studies which have investigated cCAFs in blood samples [34,35,36], and there is a lack of preclinical studies showing cCAF formation. In this study, our microsieve technology platform was able to successfully capture and enrich both CTCs and cCAFs in blood from tumor-bearing mice. In line with previous studies, we used pan-CK as a marker for CTC detection and tested four markers (vimentin, α-SMA, FAP and ITGA5) to detect cCAFs. However, vimentin failed to reveal differences at either the mRNA level or in the cell-staining setting, possibly due to its non-specificity between MDA-MB-231 breast cancer cells and fibroblasts. The same result was reported by Ao et al., indicating that, apart from fibroblasts, vimentin can also be expressed by lymphocytes and breast cancer cells [35]. Although α-SMA was the first identified CAF marker, its expression has recently been shown to be more specific to a subpopulation of CAFs, i.e., myofibroblastic CAFs (myCAFs), than all CAFs [37]. FAP is another frequently used marker for CAF identification; however, a study by Tchou et al. indicated that FAP can also be expressed by macrophages [38]. Liao et al. also reported the presence of the soluble form of FAP in human blood [39], which may explain the unspecific binding observed in our blood sample with FAP antibody (Appendix A). Previously, we found that the expression of ITGA5 (also known as CD49e) was specifically increased in activated fibroblasts in a pancreatic tumor model [26] (Appendix A). A similar result was confirmed by a recent study by Hussain and colleagues, in which they demonstrated that ITGA5 can be used as a new CAF marker in ovarian cancers [40]. It is worth mentioning that they found two different CAF populations with high and low FAP expression, and both were positive for ITGA5 [40], which suggests that ITGA5 may identify more CAF types than FAP. These data encouraged us to investigate ITGA5 as a potential marker for cCAF identification, which was confirmed in this study.

Although identification of CTCs is beneficial for evaluation of tumor prognosis in clinical studies [35,41,42], in our study, tumors developed with only cancer cells alone (T-alone) exhibited slow tumor growth compared to tumors developed with coinjection of cancer cells and fibroblasts (T + F). However, the enumeration and incidence of CTC detection in these groups was not much different (Figure 3). On the other hand, higher rates of enumeration (3–15 cCAFs/100 μL vs. <2 cCAFs/100 μL) and incidence (6/6 vs. 2/6) were found for cCAFs in the stroma-rich T + F group vs. the T-alone group, respectively. The presence of cCAFs in the T-alone group might be the result of the epithelial–mesenchymal transition (EMT) of cancer cells or recruitment of host fibroblasts in the tumor microenvironment [43] (Appendix A). These data indicate that tumors with more stromata (T + F) exhibit faster tumor growth and tend to have more cCAFs in blood. Another study showed significantly higher cCAF counts in metastatic colorectal cancer patients compared to localized prostate cancer patients [35]. In this study, cCAFs but not CTCs were found in three patients who had metastasis, indicating the significance of cCAF detection, in addition to CTCs, with respect to metastasis prognosis. In line with this research, we found that some mice were detected with a high number of cCAFs but without any CTCs detected (Figure 3B). Ortiz-Otero et al. recently reported that cCAF levels are positively correlated with worse overall survival of patients [34]. Taken together, these data suggest the potential of cCAFs as an indicator to predict cancer progression, although further preclinical and clinical studies in different cancer types are required to confirm this phenomenon.

Previous studies have shown that chemotherapy can enhance metastasis formation [30,31,32]. A clinical study also showed that different chemotherapies induce the release of CTCs and CAFs in metastatic breast cancer patients [34]. However, to the best of our knowledge, there is a lack of preclinical studies investigating the effect of chemotherapy on CTC and cCAFs. In line with the clinical data, we found more CTCs and cCAFs in chemotherapy-treated mice (Figure 4). The counts of CTCs and cCAFs varied, irrespective of tumor size or weight, in chemotherapy-treated mice, but the detection incidence was much higher compared to untreated mice (100% vs. 33.3–66.7%). Previous studies have shown that chemotherapeutics such as docetaxel and paclitaxel in cancer-bearing mice and patients with metastatic cancers increase the dissemination of cancer cells into blood or lymphatic vessels [32,34,44]. One hypothesis is that chemotherapeutic agents can increase the permeability of vasculature, thereby facilitating the mobilization of cancer cells into the circulation [34]. This might be the reason for the increased number of cCAFs. Furthermore, we observed homotypic clusters of CTCs and cCAFs, as well as heterotypic cCAF–CTC clusters in the chemotherapy group. These data are in line with previous clinical studies in which such clusters were reported [34,35,45], confirming our data and indicating that cCAFs might travel along with CTCs to support their survival in host organs to form distant metastasis [13,46]. Ao et al. demonstrated the detection of clustering of cCAFs in patients with cancer for the first time and revealed that such clusters were correlated with cancer metastasis [35]. Clusters of CAF cells can protect and support the survival of CTCs during circulation and facilitate the settling of cancer cells in distant sites; thus, heterotypic cCAF–CTC clusters have a greater ability to survive during circulation. In a recent study by Sharma et al. cCAF–CTC heterotypic clusters were also detected in patients and mouse models, in which they found that heterotypic clusters have higher metastatic potential than homotypic CTC clusters [45]. Taken together, these data suggest that identification of not only CTCs but also cCAFs, especially heterotypic CTC clusters with cCAFs or other cells, is important in monitoring cancer progression.

## 5. Conclusions

In conclusion, this study shows the detection of cCAFs based on a novel biomarker, ITGA5, in blood in an orthotopic breast tumor xenograft model. Interestingly, stroma-rich tumors showed higher rates and numbers of cCAFs compared to stroma-poor tumors. Furthermore, treatment with chemotherapy induced the release of both CTCs and cCAFs in blood, which also tend to form clusters. Therefore, detection of cCAFs using ITGA5 as a marker, together with the detection of CTCs, might provide cues to monitor tumor progression and the effects of chemotherapy on metastatic dissemination.

## Figures and Tables

**Figure 1 cancers-15-01127-f001:**
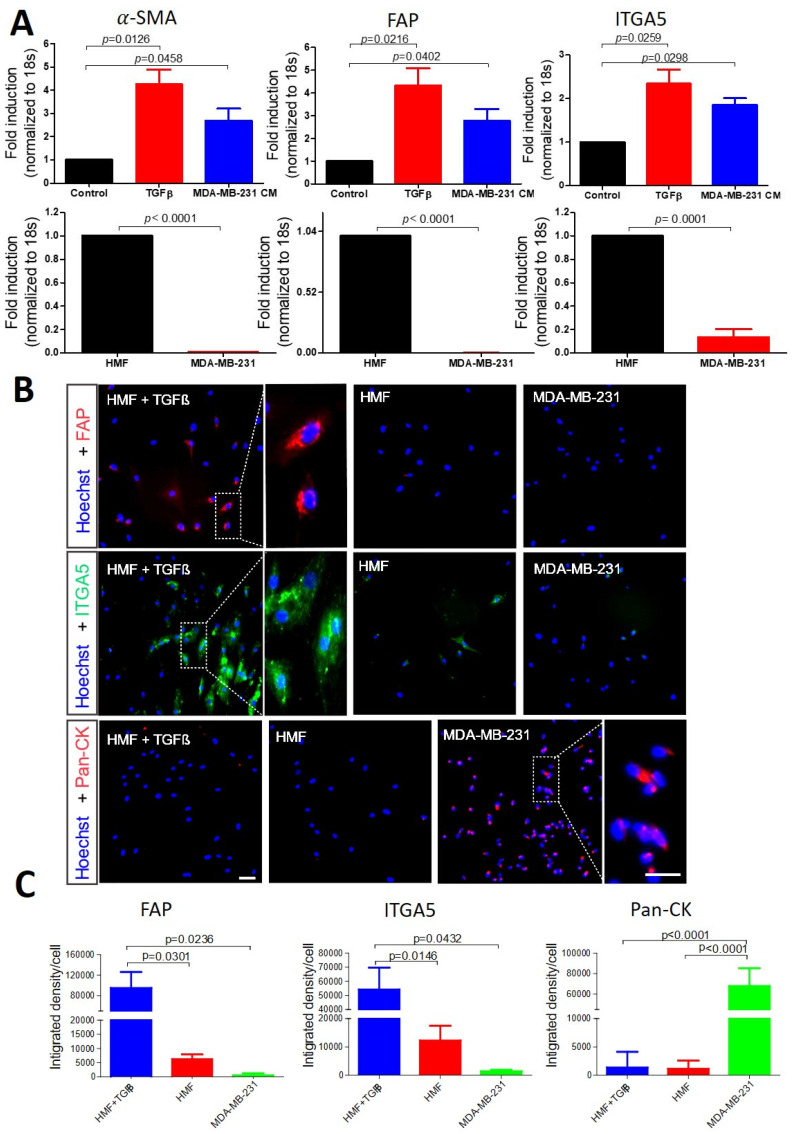
Biomarkers to identify circulating CAFs and tumor cells. (**A**) Gene expression analysis using qPCR for α-SMA, FAP and ITGA5 in activated and non-activated HMFs with TGFβ or conditional medium collected from cancer cells and human MDA-MB-231 cancer cells. (**B**) Immunofluorescence staining for FAP (red), ITGA5 (green) and pan-CK (red) in non-activated and TGFβ-activated HMFs, as well as MDA-MB-231 breast tumor cells; Hoechst (blue) represents nuclear staining. Scale bar, 100 µm. (**C**) Bar graphs showing the quantitative analyses of each staining (**B**) analyzed using Image J software. Data are presented as the mean + SEM of three independent experiments. Statistical analyses show differences between groups as *p*-values using two-sided unpaired Student’s *t* test.

**Figure 2 cancers-15-01127-f002:**
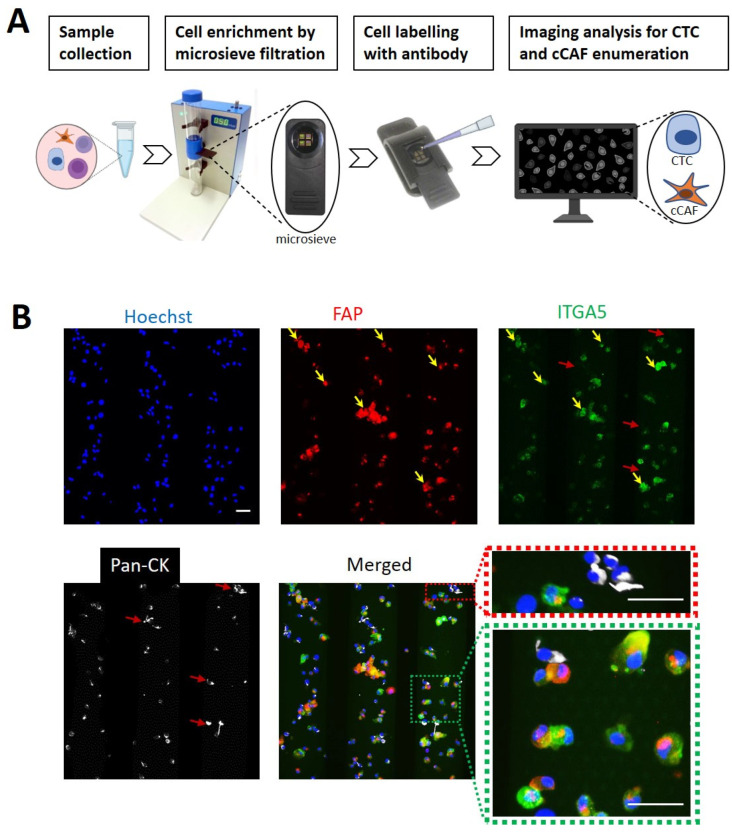
Microsieve filtration platform for circulating cell capture, enrichment and identification. (**A**) Schematic illustration of the workflow for cCAF and CTC capture using the microsieve filtration platform. (**B**) In vitro activated HMFs mixed with tumor cells for selected target staining. Yellow arrows indicate FAP (red) and ITGA5 (green) costaining on HMF, while the white signal indicates pan-CK specifically on MDA-MB-231 cells without any costaining on HMFs (red arrows). Scale bar, 40 μm.

**Figure 3 cancers-15-01127-f003:**
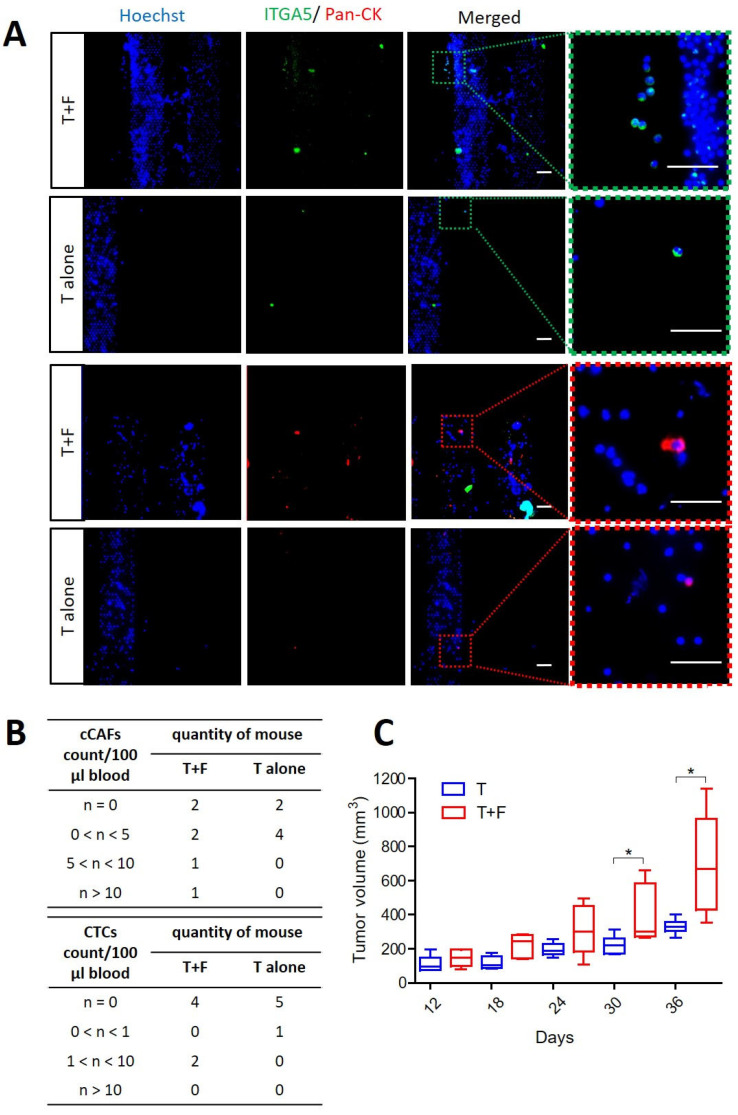
Effect of CAFs on tumor growth and CTC and cCAF formation in vivo. Tumor-bearing mice with breast tumors formed with (T + F) or without fibroblasts (T alone) showed differences in circulating cell detection and tumor growth. (**A**) cCAFs and CTCs were detected in blood from human MDA-MB-231 tumor-bearing mice. ITGA5 (green) indicates the cCAF population, while pan-CK (red) indicates CTCs. (**B**) Tables listing the levels of cCAFs and CTCs detected from tumor-bearing mice. (**C**) Significantly larger tumor size was observed in T + F mice compared to T-alone mice (* *p* = 0.0424 on day 30, * *p* = 0.0169 on day 36) using a two-sided unpaired *t* test. Data are presented as the mean ± SEM, N = 6. Scale bar, 40 μm.

**Figure 4 cancers-15-01127-f004:**
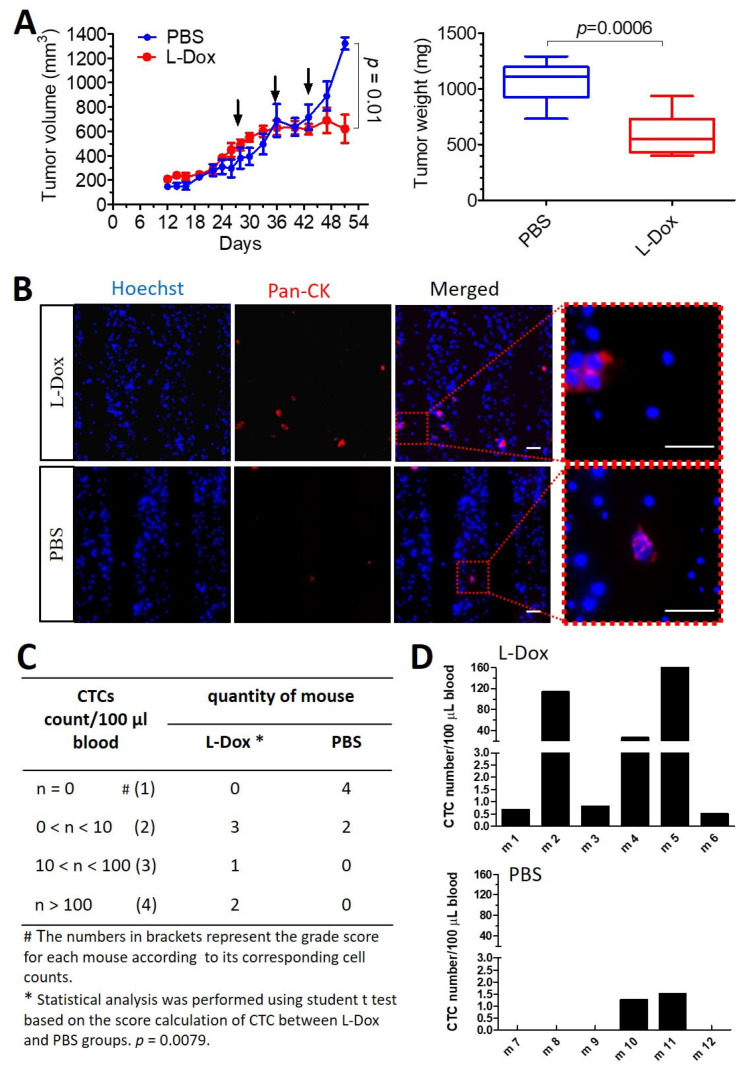
Chemotherapy reduced tumor growth but induce CTCs in vivo. (**A**) Mice treated with L-Dox at a dose of 5 mg/kg once/week for 3 weeks (red arrows indicate L-Dox administration) showed a significant suppression of tumor growth and tumor weight. Data are presented as mean + SEM, N = 6. *p*-value was calculated with a two-sided unpaired Student’s *t* test. (**B**) CTCs were captured and identified in both groups of mice. Scale bar, 40 μm. (**C**) Table listing the levels of CTCs detected in mice treated with L-Dox or PBS. Statistical analysis was performed based on the grading score system according to the CTC count of each mouse (score 1: 0 CTCs; score 2: 0–10 CTCs; score 3: 10–100 CTCs; score 4: >100 CTCs) using a two-sided unpaired Student’s *t* test, showing significant differences between the L-Dox and PBS groups; *p* = 0.0079. (**D**) CTC numeration for each mouse in the L-Dox- and PBS-treated groups.

**Figure 5 cancers-15-01127-f005:**
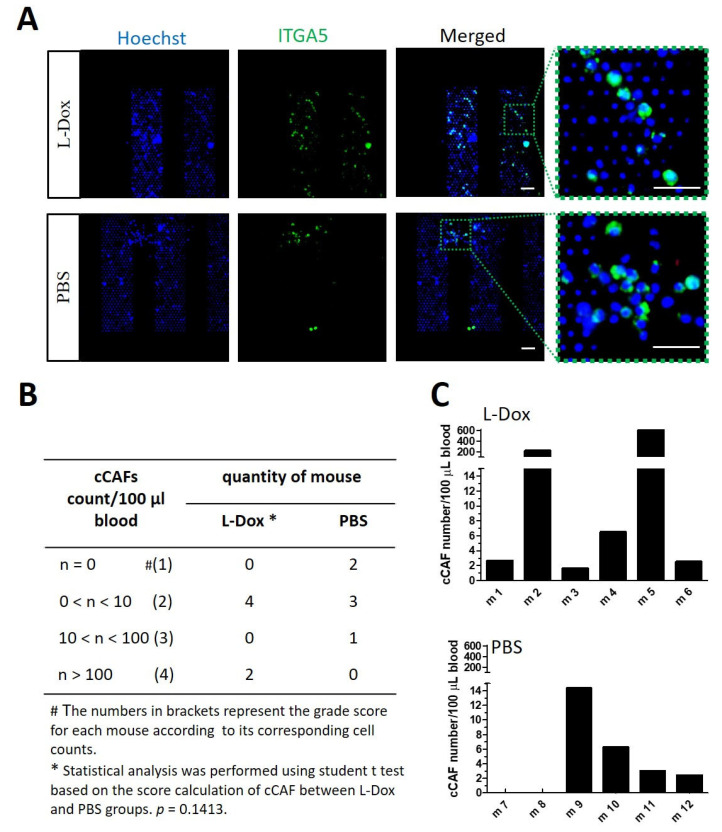
Chemotherapy treatment induced cCAFs in vivo. (**A**) Immunostaining of ITGA5 (green) in blood from mice treated with L-Dox (5 mg/kg once/week for 3 weeks) to detect cCAFs compared with untreated mice. Scale bar, 40 μm. (**B**) Table listing the levels of cCAFs detected in mice treated with L-Dox or PBS. Statistics of (**C**) cCAF numeration for each mouse in the L-Dox- and PBS-treated groups. Statistical analysis was performed based on the grading score system according to the cCAF count of each mouse (score 1: 0 cCAFs; score 2: 0–10 cCAFs; score 3: 10–100 cCAFs; score 4: >100 cCAF)s using a two-sided unpaired Student’s *t* test, showing differences with *p*-values of 0.1413 between the L-Dox and PBS groups.

**Figure 6 cancers-15-01127-f006:**
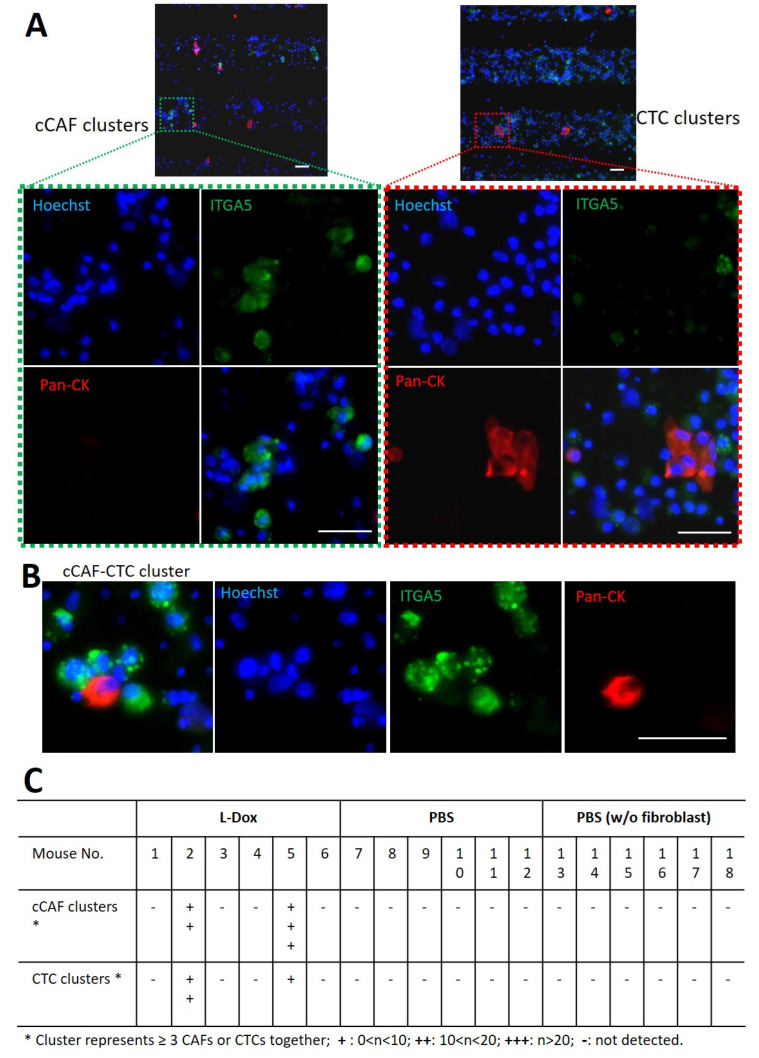
Clusters of CTC and cCAF detected in blood from mice with breast tumors. (**A**) Representative pictures of a cCAF cluster and a CTC cluster captured in blood. (**B**) Double-fluorescence images showing clusters of CTCs and cCAFs. (**C**) Table listing the overall clustering identification of each mouse in different treatment groups, showing that clusters of CTCs and cCAFs were detected in L-Dox-treated mice only. Scale bar, 40 μm.

## Data Availability

The data presented in this study are available in this article (and Appendix A).

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
