# Peer review of "In Vivo Detection of Circulating Cancer-Associated Fibroblasts in Breast Tumor Mouse Xenograft: Impact of Tumor Stroma and Chemotherapy"

_cancers, 2023, doi:10.3390/cancers15041127_

Round 1
Reviewer 1 Report
This is an interesting study foxusing on the role of activated fibroblasts in the development of metastasis of breast cancer, in an experimental model. Authors could consider the following points:
1. Results Selection of biomarkers for cancer associated fibroblasts and cancer cells identification. Do co-cultures of HMF with cancer cells activate HMFs (induce ITGA5) ?
2. Figure 1A: 1A shows that aSMA and FAP mRNA increase after exposure to TGFb. Why aSMA did not increase in fluorescence? Western blot may help to check protein levels. Please also check the '!-SMA (?)' in fig 1A.
3. Results. CAFs promote tumor growth and support dissemination of CTCs and cCAFs.
Why CAFs were detected in the blood of mice implanted with cancer cells without HMFs. Did naive mice without tumor implantation have activated circulating fibroblasts? Presumably these correspond to mouse fibroblasts that became activated by cancer cells? If so, does the tumor growing after cancer cell implantation attract mouse fibroblasts? This should be examined with immunofluoresecence staining of the tumor mass.
4. Were mice treated with LDOX implanted with mixed cancer cells and human fibroblasts or with cancer cells only?
5. The assessment of circulating activated HMFs and clusters with CCs is important. However, the study does not provide evidence that these clusters form metastasis and that HMFs are included in the tumoral mass. This is a disadvantage of the study.
Author Response
We would like to thank the reviewer for the valuable comments. We have sincerely considered the comments and addressed them in the revised version of the manuscript. Below are the point-to-point reply to the comments. We have also made changes in the manuscript which are highlighted in red color.
- ResultsSelection of biomarkers for cancer associated fibroblasts and cancer cells identification. Do co-cultures of HMF with cancer cells activate HMFs (induce ITGA5)?
Answer:Thank you for your comment. Yes, we do see an activation of HMF and induction of ITGA5 when we add conditioned media from cancer cells to HMF. We used conditioned media from cancer cells instead of co-culture because this provides the effect of cancer cells on HMF without physical interaction. In this way, we can examine the changes in gene expression in a cleanest way. These data are presented in figure 1A.
- Figure 1A: 1A shows that aSMA and FAP mRNA increase after exposure to TGFb. Why aSMA did not increase in fluorescence? Western blot may help to check protein levels. Please also check the a-SMA (?)' in fig 1A.
Answer: Thank you for your comment. We have now performed the western blot analysis for a-SMA and observed the increased expression after TGF-b activation. We added the result in the manuscript as Supplementary figure 1A, B and text on page 6.
- Results. CAFs promote tumor growth and support dissemination of CTCs and cCAFs.
Why CAFs were detected in the blood of mice implanted with cancer cells without HMFs. Did naive mice without tumor implantation have activated circulating fibroblasts? Presumably these correspond to mouse fibroblasts that became activated by cancer cells? If so, does the tumor growing after cancer cell implantation attract mouse fibroblasts? This should be examined with immunofluoresecence staining of the tumor mass.
Answer: This is true that tumor cells can attract the fibroblasts from the host tissue and therefore we see stroma in tumors only injected with tumor cells. We have not performed a-SMA staining on the tumor tissue because we have the optimized staining method with a mouse derived anti-SMA antibody which gives background due to the use of mouse antibody on mouse tissue. We have therefore stained the tumor tissue with anti-collagen 1α1, and found that the mice with only cancer cells also show collagen staining, confirming that fibroblasts are there in the tumor to produce collagen. We add this part as a Supplementary figure 4 and page 18. Besides, our previous study on pancreatic cancer demonstrated that the mice implanted with cancer cells only (panc-1 cell) also showed a-sma and collagen positive in tumor (Ref: doi: 10.1126/sciadv.aax2770). These indicate that cancer cells are able to recruit the fibroblasts from the host and support the tumor growth.
- Were mice treated with LDOX implanted with mixed cancer cells and human fibroblasts or with cancer cells only?
Answer:L-DOX treated mice were implanted with mixed cancer cells and human fibroblasts. We have made it more clear in the description of the revised manuscript on page 5.
- The assessment of circulating activated HMFs and clusters with CCs is important. However, the study does not provide evidence that these clusters form metastasis and that HMFs are included in the tumoral mass. This is a disadvantage of the study.
Answer: We thank the reviewer for the comments. We did not find metastasis formation in these mice. However, there have been studies in literature which show that CTCs migrate along with CAFs as cell aggregates which facilitates tumor cell survival and metastasis formation (Ref: https://doi.org/10.1073/pnas.1016234107; https://doi.org/10.1084/jem.20180765; https://doi.org/10.18632/oncotarget.27510). We have cited these papers and discussed this on page 19.
Reviewer 2 Report
This is well structured study including study design and results.
Thanks.
Author Response
We thank the reviewer for the appreciation.
Reviewer 3 Report
The study is interest to the literature and bring new approach to cancer microenvironment. I recommend the ms for publication.
Author Response

(The authors gave the same response as above.)

Reviewer 4 Report
In this manuscript, the authors claimed that they identified ITGA5 as a cCAF biomarker, and found correlations between HMF and cCAF in their experimental settings. However, the experimental design has flaws, and the biological story is not complete.
1. The authors claimed that ITGA5 is a cCAF biomarker. However, they only performed experiments with one human breast cancer cell line MDA-MB-231. It’s possible that the upregulation of ITGA5 only exists in this cell line but not in other breast cancer cell lines. The authors should check more different cell lines to prove that ITGA5 is a broad cCAF biomarker in human breast cancer.
2. The authors found that the ITGA5+ cCAFs were detected more in T+F group mice than T alone group mice. However, the authors cannot exclude the possibility that this phenotype is caused by a technical issue, in which the HMFs were leaked into the circulation during the orthotopic injection. To confirm the appearance of ITGA5+ cCAF is correlated with tumor development, the authors should add a negative control group, in which they inject HMF only without tumor cells into mice and check the cCAF level.
3. In Fig 4 and Fig 5, the authors should give statistical analyses of the CTCs and cCAFs in L-DOX vs. PBS groups.
4. The phenotype described in Fig 6 was very rarely detected in the in vivo experiment. Therefore, it’s difficult to evaluate whether this phenotype is indeed biologically meaningful.
5. In Fig 3, since the cCAFs and CTCs are very rare populations in the blood, their detection may be easily interrupted by false positive signals (for example, non-specific binding of antibodies to other blood components). The authors should add a negative control group, in which they collect blood from healthy littermate mice without tumor cells or HMFs injection and stain with ITGA5 and pan-CK antibodies.
Author Response
We would like to thank the reviewer for the valuable comments. We have sincerely considered the comments and addressed them in the revised version of the manuscript. Below are the point-to-point reply to the comments. We have also made changes in the manuscript which are highlighted in red color.
- The authors claimed that ITGA5 is a cCAF biomarker. However, they only performed experiments with one human breast cancer cell line MDA-MB-231. It’s possible that the upregulation of ITGA5 only exists in this cell line but not in other breast cancer cell lines. The authors should check more different cell lines to prove that ITGA5 is a broad cCAF biomarker in human breast cancer.
Answer:We would like to emphasize that ITGA5 was overexpressed in HMF but not in breast tumor cell line. HMFs are healthy primary human mammary fibroblasts and we show that the expression of ITGA5 is induced in HMFs either via TGF-b (as a common growth factor for fibroblast activation) or conditioned media. Previously we have also examined ITGA5 expression in human healthy pancreatic stellate cells (becoming CAF after TGFb activation). We also observed that ITGA5 was upregulated in these cells but not in pancreatic cancer cells. We have added these data as supplementary figure 3 in the manuscript and p17-18. Additionally, our previous study has also proved this (Ref: doi: 10.1126/sciadv.aax2770. Hence, ITGA5 can be used as a biomarker for cCAF.
- The authors found that the ITGA5+ cCAFs were detected more in T+F group mice than T alone group mice. However, the authors cannot exclude the possibility that this phenotype is caused by a technical issue, in which the HMFs were leaked into the circulation during the orthotopic injection. To confirm the appearance of ITGA5+ cCAF is correlated with tumor development, the authors should add a negative control group, in which they inject HMF only without tumor cells into mice and check the cCAF level.
Answer:This is highly unlikely that HMFs were leaked during the injection and then circulated for weeks in blood. One important observation is that cCAFs were also detected in the blood of mouse which had only cancer cell injection but no HMFs. This suggests that cCAFs are recruited to the tumors from the host tissue and then released in the blood. In case of co-injection, if there was a leakage from cell injection (mixture of cancer cells and fibroblasts), then all mice positive for cCAFs should have also been positive for CTCs. However, only 2 mice were positive for CTCs but 4 mice for cCAFs (Figure 3B). This observation indicates that the detection of cCAFs was not a technical issue.
- In Fig 4 and Fig 5, the authors should give statistical analyses of the CTCs and cCAFs in L-DOX vs. PBS groups.
Answer:Because there is a high variation in the CTC and cCAFs numbers, we have graded the counts based on the table shown in the figures and performed statistics. The grading score was given according to the cCAF count in each mouse (score 1: no cells; score 2: 0-10 cells; score 3: 10-100 cells; score 4: >100 cells for both cCAFs and CTCs). We found that the difference between CTC count is statistically significant (p=0.0079) while for cCAFs it is not significant but p=0.14. These data are included in the revised manuscript in respective figures (4 and 5) and on page 13-14.
- The phenotype described in Fig 6 was very rarely detected in the in vivo experiment. Therefore, it’s difficult to evaluate whether this phenotype is indeed biologically meaningful.
Answer:In comparison with single CTC, the clusters are relatively rare to be detected. However, detection of clusters of CTC-cCAF has been reported in several studies (Ref: doi: 10.1158/0008-5472.CAN-15-1633; doi.org/10.1007/s10549-021-06299-0; doi.org/10.1186/s12885-020-07376-1.) These studies indicated that the heterotypic clusters of CTC-CAF facilitate metastasis formation. Besides, for homotypic cluster of CTCs detection can be traced back to the year of 1954 (Ref: https://doi.org/10.1002/1097-0142(195403)7:2<215::AID-CNCR2820070203>3.0.CO;2-6 ). To date more evidences have demonstrated the detection of CTC clumps in patients with various cancers, showing the positive correlation between CTC clusters and easy formation of metastases. Hence, finding the clutsers of CTC or cCAF and CTC-CAF in our study, we think, is biologically meaningful to present the information that chemotherapy may also contribute the release of clusters of CTC/CAF.
- In Fig 3, since the cCAFs and CTCs are very rare populations in the blood, their detection may be easily interrupted by false positive signals (for example, non-specific binding of antibodies to other blood components). The authors should add a negative control group, in which they collect blood from healthy littermate mice without tumor cells or HMFs injection and stain with ITGA5 and pan-CK antibodies.
Answer:To avoid false positive signals, we first optimized the method using fresh blood without tumor cells and confirmed that there was no artifact of false signal and cluster. In supplementary figure 2, we provide an example in which we found a reaction with blood and false positive signal which was removed from the analysis. Therefore, we are confident with our methodology that there are no false positive signals.
Round 2
Reviewer 4 Report
The explanation and revision are satisfying.